# Utilizing AgNPt-SALDI to Classify Edible Oils by Multivariate Statistics of Triacylglycerol Profile

**DOI:** 10.3390/molecules26195880

**Published:** 2021-09-28

**Authors:** Tzu-Ling Yang, Cheng-Liang Huang, Chu-Ping Lee

**Affiliations:** 1Department of Applied Chemistry, National Chiayi University, Chiayi City 60004, Taiwan; facebook6119@gmail.com (T.-L.Y.); clhuang@mail.ncyu.edu.tw (C.-L.H.); 2Department of Chemistry, Fu Jen Catholic University, New Taipei City 24205, Taiwan

**Keywords:** MALDI, SALDI, edible oil, nanoparticles, principal component analysis, fingerprints, multivariate statistics

## Abstract

Edible oils are valuable sources of nutrients, and their classification is necessary to ensure high quality, which is essential to food safety. This study reports the establishment of a rapid and straightforward SALDI-TOF MS platform used to detect triacylglycerol (TAG) in various edible oils. Silver nanoplates (AgNPts) were used to optimize the SALDI samples for high sensitivity and reproducibility of TAG signals. TAG fingerprints were combined with multivariate statistics to identify the critical features of edible oil discrimination. Eleven various edible oils were discriminated using principal component analysis (PCA). The results suggested the creation of a robust platform that can examine food adulteration and food fraud, potentially ensuring high-quality foods and agricultural products.

## 1. Introduction

Edible oils are essential and provide fundamental nutrients for daily life, such as fatty acids, triacylglycerol, and fat-soluble vitamins (A, D, E, and K) [1,2]. Edible oil extracts encompass olive, palm, soybean, canola, corn, and other plant sources. In recent years, food safety has been compromised by the fraud and adulteration of oils, raising health concerns [3,4]. Additionally, the use of compromised oils has a severe economic cost. The comprehensive determination of high-quality, all-natural edible oils would prevent this rampant misconduct.

Various methods, such as infrared spectroscopy [5], Raman spectroscopy [6], gas chromatography [7], and mass spectrometry (MS) [8,9], are utilized for edible oil analysis. Electrospray ionization (ESI)-MS [10,11] is the primary tool used to characterize edible oils. There are many high-quality studies performed utilizing this methodology. Based on fatty acid and triacylglycerol (TAG) fingerprints, Li et al. [12] reported the characterization of edible vegetable oils using PCA. Piñero et al. [13] employed ESI-MS analysis to define the quality of 30 olive oil samples, which were classified as extra virgin olive oil, virgin olive oil, and lower quality oils. These studies required time-consuming pretreatments, such as extraction before ESI-MS detection. ESI-MS can provide detailed information for the identification of edible oils from the same species but uses various production processes. The current study focuses on the classification of edible oils from plant sources with independent production processes to examine their adulteration. A rapid and robust method, matrix-assisted laser desorption/ionization (MALDI), is suitable for oil sample analysis [14,15,16,17,18]. In a 2015 study, Ng et al. [19] established a preliminary MALDI-MS spectral database for the classification of edible oils using a hierarchical clustering analysis (HCA). In a 2018 study, Ng et al. [20] measured 30 types of edible oil using MALDI-MS. PCA classified oils into eight groups. In a 2019 study conducted by Kuo et al. [21], MALDI–time-of-flight (TOF)-MS was used to analyze edible oil samples; the fingerprints of TAGs were counted through similarity analysis and PCA was used for classification. However, this methodology is hindered by the lack of matrix/sample homogeneity (“sweet spot”) during matrix crystallization in the MALDI sample preparation. This problem causes low reproducibility and large fluctuations in the intensities of signals, making the quantitative analysis of oils difficult.

Interestingly, nanomaterials can be employed to improve MALDI signal reproducibility; this technique is called surface-assisted laser desorption/ionization (SALDI) [22,23,24]. SALDI sample preparation does not require the crystallization of the matrix and analyte. Hence, the sample morphology is more homogenous. A wide variety of materials have been developed to assist SALDI detection, including, among others, metallic nanoparticles, metallic oxide nanoparticles, and materials with porous surfaces. Various nanomaterials have highly selective signal enhancement; therefore, different nanomaterials are suitable for detecting different analytes. For example, silver nanoparticles (AgNPs) can be employed to detect perfluorinated sulfonic acid in environment pollution [25] and TAG in brain tissue [26]. In contrast, gold nanoparticles can be used to detect aminothiols during metabolic disease diagnosis [27] and mercury telluride nanomaterials (HgTe) for glycoconjugate detection [28]. Palladium nanoparticles were used to assist SALDI detection when identifying the characteristics of various edible oils, especially fatty acids and TAGs [29].

In the present study, AgNPs are used to assist with the detection of TAGs in edible oil. The properties of AgNPs are related to their shape, size, refractive index, and surface modification, which are controllable through numerous synthesis methods [30,31,32]. Different forms of AgNPs were prepared for use in a SALDI sample, and their performance was compared. Sample preparation optimization can be applied to SALDI-MS to detect the characteristics of edible oil and characterize TAGs. To confirm that the TAG profile that was obtained was reliable, intra-day and inter-day experiments were conducted. The discrimination of TAG profiles in various edible oils was performed using multivariate statistics.

## 2. Results

### 2.1. Use of AgNPs in Edible Oil Detection Using MALDI-MS

Various synthesis methods have been developed to control the properties of AgNPs, such as their shape, size, surface modifications, and refractive index [30,31,32]. Nanoparticles as diverse as AgNPts [32] and near-spherical AgNPs (decahedron and icosahedron forms) [31,33] were used to assist SALDI, which may affect the morphology of the SALDI sample. Figure 1A shows the absorption spectrum of the aqueous AgNP mixture. The blue line is the spectrum of the AgNPt solution and contains out-of- and in-plane quadrupole surface plasmon resonance (SPR) bands in the ranges of 320–350 and 350–550 nm, respectively. The green line is the spectrum of the D-AgNP solution and spans 330–470 nm. The orange line is the spectrum of the I-AgNP solution and spans 425–550 nm. The appreciable absorption (extinction) of the AgNPts was observed at a wavelength of 355 nm of the third-harmonic generation Nd:YAG laser, which was employed to mimic commercial MALDI-MS laser conditions. We then obtained TEM images of the various AgNPs (Figure 1B). The majority of the AgNPts had triangular plate nanostructures with edge lengths of 50–80 nm. The diameters of the D-AgNPs and I-AgNPs were 85–105 and 35–45 nm, respectively, and both forms were nearly spherical. Nanostructure-based surfaces can be used to improve ionization efficiency because of their photoabsorption and photothermal abilities. Studies have suggested that the SPR of nanomaterials generates both hot-electron transfer and enhanced electromagnetic effects, corresponding with the SALDI ionization mechanism [32,34,35,36,37]. We thus examined the ability of the nanostructures to be used as assisting materials in SALDI-MS. The AgNPts produced the largest TAG ion intensity, increasing eight times higher than when the D-AgNPs or I-AgNPs were under the same laser energy (Figure 1C). However, these experimental results were in disagreement with the extinction coefficient of the materials, which followed the order I-AgNPs > AgNPts > D-AgNPs. Extinction within the desorption/ionization process is crucial for absorption at 355 nm. Additionally, the AgNP structure is essential to the SALDI process, and the triangular shape would result in various electromagnetic effects and affect ion generation [35,36,37]. Among the three AgNPs investigated, the AgNPts were discovered to have sharp corners, whereas the D-AgNPs and I-AgNPs were nearly spherical (Figure 1B). The AgNPts were used to assist material in optimizing the concentration of edible oil for TAG analysis (Appendix A). The TAG signals were obtained from edible oils at different concentrations, showing significant differences. Stronger TAG signals were detected in edible oil samples at the concentration of 1.0 μL mL^−1^ than in samples at the concentrations of 2 to 10 μL mL^−1^. We thus used AgNPts as the assisting material and an edible oil concentration of 1.0 μL mL^−1^ as the analyte for further experiments in this work.

The TAG signal intensity was measured when the AgNPts, CHCA, and 2,5-DHB were employed under three laser energy conditions (Appendix A). For all laser powers, less background noise was observed when the AgNPts were used as an assisting material. The observed peaks at *m/z* 881.7 and 907.7 corresponded to the peaks of sodiated TAG 52:2 and TAG 54:3 ions, respectively. The AgNPts and 2,5-DHB produced similar TAG signal intensities under various laser energies. Signal reproducibility is a drawback of MALDI-MS. To demonstrate the reproducibility of the detection made using the AgNPts, we compared the results obtained using the AgNPts with those obtained using 2,5-DHB and CHCA under three sample preparations. For the comparison of these matrices and assisting material, the CHCA provided a homogenous sample morphology and high sensitivity. However, it produced a higher background interference in this mass range (lower *m*/*z* 1000), indicating that the CHCA is not suitable for the observation of TAG ion from edible oils. The TAG ion intensities obtained using the AgNPts and 2,5-DHB were not considerably different. Subsequently, we examined the reproducibility of the AgNPt and 2,5-DHB results using a thin-layer preparation; using the AgNPts as the matrix resulted in a higher reproducibility (reproducibility standard deviation (RSD) 41.67%) than using the 2,5-DHB (RSD 62.20%). In MALDI, sample preparation is a crucial parameter affecting ion intensity. Various sample preparations of the AgNPts (Figure 2) were examined by considering the relative standard deviation of the TAG signals (TAG 52:2 and TAG 54:3). For comparison, we examined thin-layer, dried-droplet, and solution spray sample preparations. The solution spray method yielded a lower RSD (22.42%) than the thin-layer method (RSD 41.67%) and dried-droplet method (RSD 52.04%). Among these methods, the solution spray method provided the most homogenous sample.

### 2.2. Fingerprint-Based Quality Control

We performed a temporal study to evaluate the stability of TAG 52:2 and TAG 54:3 in olive oil. TAG 52:2 and TAG 54:3 signals were collected from 25 random positions within each sample, and 5 samples were examined within 1 day. Reproducibility was evaluated using the RSD, which was calculated from the variation in ion intensity. Figure 3A illustrates the intra-sample and intra-day results. The RSDs were in the range of 11.80–16.23% for intra-sample reproducibility and 8.28% for intra-day reproducibility. To examine the signal reproducibility, we performed the experiments on 5 individual days. Figure 3B presents the intra-day and inter-day findings. The intra-day RSDs were in the range of 5.82%–8.28%. The TAG signals from the five individual experiments were calculated; the inter-day RSD was determined to be 13.21%; and the TAG signals were repeated in the intra-sample, intra-day, and inter-day experiments.

### 2.3. Classification of Edible Oils by Multivariate Statistics

To further demonstrate the capability of the AgNPt solution spray method for TAG detection, the solution was applied to prepared SALDI samples to analyze eleven edible oils (Figure 4). Various TAG species were observed as [M + Na]^+^ ions (Appendix A). Details of the mass spectrum assignments are provided in Appendix A. Three major types of TAG had 50 (*m/z* 850–870), 52 (*m/z* 870–890), and 54 (*m/z* 890–910) carbon chains with different numbers of C=C bonds. The edible oils mainly consisted of similar TAG species, such as TAG 50:1 (*m/z* 855.7), TAG 52:2 (*m/z* 881.7), TAG 54:5 (*m/z* 903.7), and TAG 54:3 (*m/z* 907.7). Figure 5A displays a heatmap visualizing the different TAG features of the eleven edible oils listed in Table 1. Using the spray AgNPt sample (Figure 5A) resulted in a superior classification compared with using the thin-layer AgNPt preparation (Appendix A). Thus, the AgNPt solution spray resulted in less fluctuation than the thin-layer AgNPt preparation.

TAG spectral fingerprints were used for statistical analysis. PCA revealed that using the spray AgNPt sample (Figure 5B) led to a more significant discrimination than using the thin-layer AgNPt method (Appendix A). Therefore, the spray AgNPt method of sample preparation offered less fluctuation and a higher discriminatory performance than the thin-layer AgNPt method. The first and third principal components (accounting for 36.5% and 16.6% of the variance, respectively) enabled the visual differentiation of avocado oil, canola oil, olive oil, rice bran oil, and sunflower oil based on their spectral fingerprints. However, with some edible oils, a higher dimension was required to differentiate them using PCA, as illustrated in Figure 5C (group 1) and Figure 5D (group 2). This suggested that these three oils in their groups (group 1: grapeseed oil, peanut oil, and soybean oil; group 2: camellia oil, flaxseed oil, and safflower oil) had similar spectral fingerprints. In a further statistical analysis, a heatmap revealed the significant difference between the TAG features of groups 1 and 2 (Appendix A). This methodology will enable us to establish a spectral database containing more numerous edible oil samples, and such a database could offer a practical and rapid classification method.

## 3. Discussion

A novel preparation using spray AgNPts as assisting material was used to analyze TAGs from edible oils. Among AgNPs, AgNPts exhibit the highest TAG ion signals relative to near-spherical AgNPs (decahedron and icosahedron) in the SALDI ionization process (a laser wavelength of 355 nm was used). It is also worth noting that the improvement of ionization efficiency is not the extinction coefficient of materials but the shape of the nanostructure. AgNPt with sharp corners may produce stronger hot-electron transfer and enhanced electromagnetic effects by SPR. The literature reported that the sharp corner of the nanostructure has a high electromagnetic field in theoretical simulation [35,36], which may enhance ionization formation [34,37]. Further experiments demonstrated that this preparation method provides a good signal, high reproducibility, and lower background interference compared to a typical matrix and a traditional preparation (e.g., thin-layer and dried-droplet process). Therefore, we used spray AgNPts to prepare the SALDI sample for edible oil analysis.

To gain more insight into these types of edible oils, a heatmap visualization provided the composition and abundance of TAG species in various edible oils. The TAG profiles of eleven edible oils could be significantly distinguished via the use of PCA. In the PCA plot, there were six edible oils separately clustered in the two groups (group 1 and group 2). This contrast was not surprising as they were mainly composed of similar TAG species. Further use of PCA plots based on the TAG features allowed us to classify three types of oils into group 1 or group 2. We believe that spray AgNPt preparation combined with multivariate statistics is a novel strategy that has the potential to classify food origins and species. Furthermore, the application can expand to the examination of food adulteration and food fraud for the quality control of foods and agricultural products.

## 4. Materials and Methods

### 4.1. Materials and Instrumentations

Silver nitrate, sodium citrate, sodium borohydride, α-cyano-4-hydroxycinnamic acid (CHCA), and 2,5-dihydroxybenzoic acid (2,5-DHB) were purchased from Sigma-Aldrich (St. Louis, MO, USA). Eleven edible vegetable oils were purchased from a local grocery store. The ultraviolet-visible absorption spectrum was employed using a Hitachi U-2800 spectrophotometer (Tokyo, Japan). Transmission electron microscopy (TEM) imaging was performed using a JEM-2100 field emission electron microscope (JEOL, Tokyo, Japan). MALDI-TOF MS was conducted using an AutoFlex III MALDI-TOF mass spectrometer (Bruker Daltonics, Bremen, Germany) with a 355 nm third harmonic Nd:YAG laser.

### 4.2. Silver Nanoparticle (AgNP) Preparation

Decahedral silver nanoparticles (D-AgNPs), icosahedral silver nanoparticles (I-AgNPs), and silver nanoplates (AgNPts) were prepared according to the procedures established in a previous study [31,32,33].

For AgNPts, a solution of sodium citrate (3.0 × 10^−2^ M, 1 mL) and a solution of silver nitrate (1.0 × 10^−2^ M, 1 mL) were added to 97 mL of pure water with rapid stirring. Then, a solution of sodium borohydride (5.0 × 10^−3^ M, 1 mL) was added dropwise to the mixture under vigorous magnetic stirring. The solution immediately turned yellow. Then, the prepared solution was irradiated with a sodium lamp (Philips, 400 W, λ_max_ = 589 nm). The typical power of the light on the solution was approximately 120 mWcm^−2^. After 90 min of irradiation, the solution turned blue, which was ascribed to AgNPt colloids.

D-AgNPs and I-AgNPs can be obtained using a seed-free, photo-assisted citrate reduction method under the irradiation of blue LEDs and violet LEDs, respectively. In a typical synthesis, 0.5 mL of sodium citrate (4.5 × 10^−1^ M) and 0.5 mL of silver nitrate (1.0 × 10^−2^ M) were mixed with 49.0 mL of pure water. The mixture was subsequently irradiated with blue LEDs (λ_max_ = 476 nm, average power 100 mWcm^−2^) or violet LEDs (λ_max_ = 405 nm, average power 100 mWcm^−2^). The colloidal solutions with an orange color and a yellow color correspond to the formations of D-AgNPs and I-AgNPs after 90 min LED irradiation, respectively.

### 4.3. Sample Preparation

Commercial edible oils were prepared in acetone at a concentration of 1.0 μL mL^−1^. The saturated CHCA and 0.1 M 2,5-DHB solution were dissolved in a solvent containing an acetonitrile–water solution (1:1, *v/v*). MALDI samples were prepared by mixing 1.0 μL of matrix—AgNPs, AgNPts, CHCA, or 2,5-DHB—with the commercial edible oils using the dried-droplet, thin-layer, or solution spray procedure. In the dried-droplet method, 1.0 μL of matrix was premixed with 1.0 μL of the edible oil solution before the mixture was deposited on a MALDI plate and vacuum dried. In the thin-layer method, 1.0 μL of matrix was deposited on the MALDI plate and vacuum dried; 1.0 μL of edible oil solution was then deposited on the plate and immediately vacuum dried. In the solution spray method, the matrix solution was sprayed on the sample plate, after which 1.0 μL of edible oil solution was deposited on the dried matrix using a pipette.

### 4.4. Data Acquisition in MALDI-TOF MS and Statistical Analysis

MALDI-TOF MS was performed using an AutoFlex III mass spectrometer (Bruker Daltonik, Bremen, Germany) in reflectron mode at a 100 Hz repetition rate, with the laser operating power at 30%. The laser spot size was set at “minimum”, ~ 80 µm in diameter. Mass spectra were obtained by accumulating 200 and 1000 individual laser shots with the “random walk” setting (10 and 25 positions; 20 shots for each position), respectively. The detector gain was set to 1739 V in the positive mode.

The collected spectral mass data .CSV files were loaded into RStudio (version 1.3.1073) [38] using the MALDIquant and MALDIquant foreign packages [34]. Data preprocessing was performed on mass spectral intensity ranging from 800 to 950 *m/z* through the following steps: quality control algorithms were employed to generate equivalent data points; baseline correction was performed using a signal-to-noise ratio (SNR) of 1, normalized by the total ion current; and peaks were aligned using the warping function. Technical replicates were averaged, and peaks were selected using a SNR of 2. The data were uniformly binned using a bin size of 0.1 Da. A data matrix was generated and exported into MetaboAnalyst 4.0 for PCA and other statistical analyses (http://metaboanalyst.ca/, accessed on 2 September 2021). The data were scaled using the standard normal variate.

## 5. Conclusions

In the present study, we used a rapid SALDI-TOF MS platform to analyze the TAG profiles of various edible oils. Using the solution spray method to prepare AgNPts with oil samples resulted in a more homogenous sample morphology and higher signal reproducibility in SALDI than in MALDI. In particular, the AgNPts with a triangular shape provided more intense TAG ion signals because of the electromagnetic effect exerted by their sharp corners and due to ion generation. TAG species (TAG 52:2 and TAG 54:3) from edible oils were observed in intra-day and inter-day experiments, and a comprehensive SALDI-TOF MS spectral database was constructed for TAG profile analysis. The TAG fingerprint PCA results revealed a clear classification of the edible oils. We developed a SALDI-TOF MS platform that is simple and robust; it can potentially be used for quality control of agricultural products and manufactured foods.

## Figures and Tables

**Figure 1 molecules-26-05880-f001:**
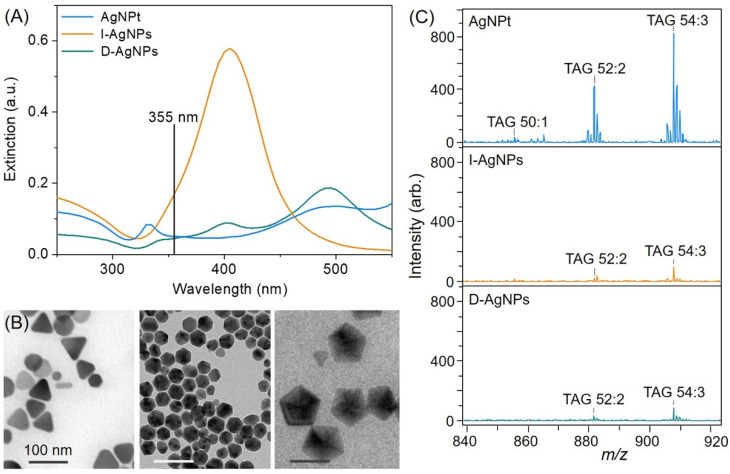
(**A**) Ultraviolet-visible absorption spectrum and (**B**) TEM images of AgNPts, I-AgNPs, and D-AgNPs. (**C**) Mass spectra of TAG profiles obtained using various AgNPs.

**Figure 2 molecules-26-05880-f002:**
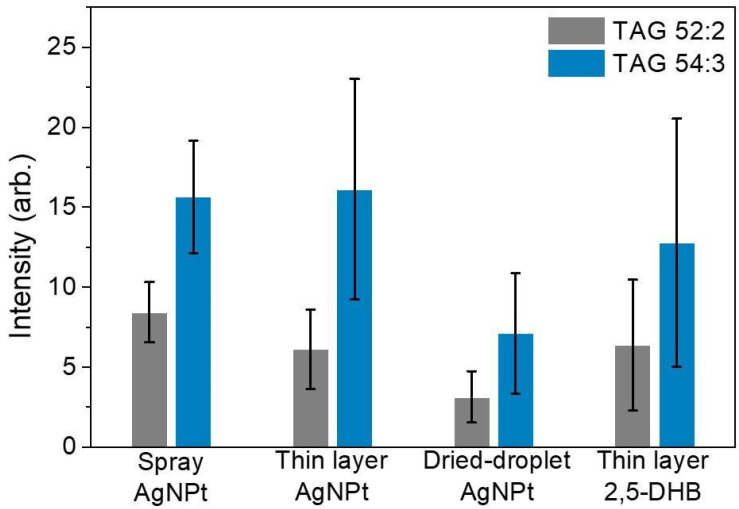
Ion intensity and error bar in the TAG signals obtained using various sample preparations: spray of AgNPt solution (RSD 22.42%), a thin layer of AgNPts (RSD 41.67%), dried droplets of AgNPts (RSD 52.04%), and a thin layer of 2,5-DHB (RSD 62.2%). The signals at *m/z* 881.7 and 907.7 correspond to [TAG 52:2+Na]^+^ and [TAG 54:3+Na]^+^, respectively. Each data bar represents the accumulation of 20 laser shots. Error bars represent the standard errors in 5 samples, including 10 spots from each sample and 20 laser shots at each sample spot. The average RSD is calculated using these two TAG signals. Gray bars represent TAG 52:2—RSDs of the TAG 52:2 signal: 22.38% (spray of AgNPt solution), 40.57% (thin layer of AgNPts), 51.15% (dried droplets of AgNPts), and 64.36% (thin layer of 2,5-DHB). Blue bars represent TAG 54:3—RSDs of the TAG 54:3 signal: 22.46% (spray of AgNPt solution), 42.76% (thin layer of AgNPts), 52.93% (dried droplets of AgNPts), and 60.03% (thin layer of 2,5-DHB).

**Figure 3 molecules-26-05880-f003:**
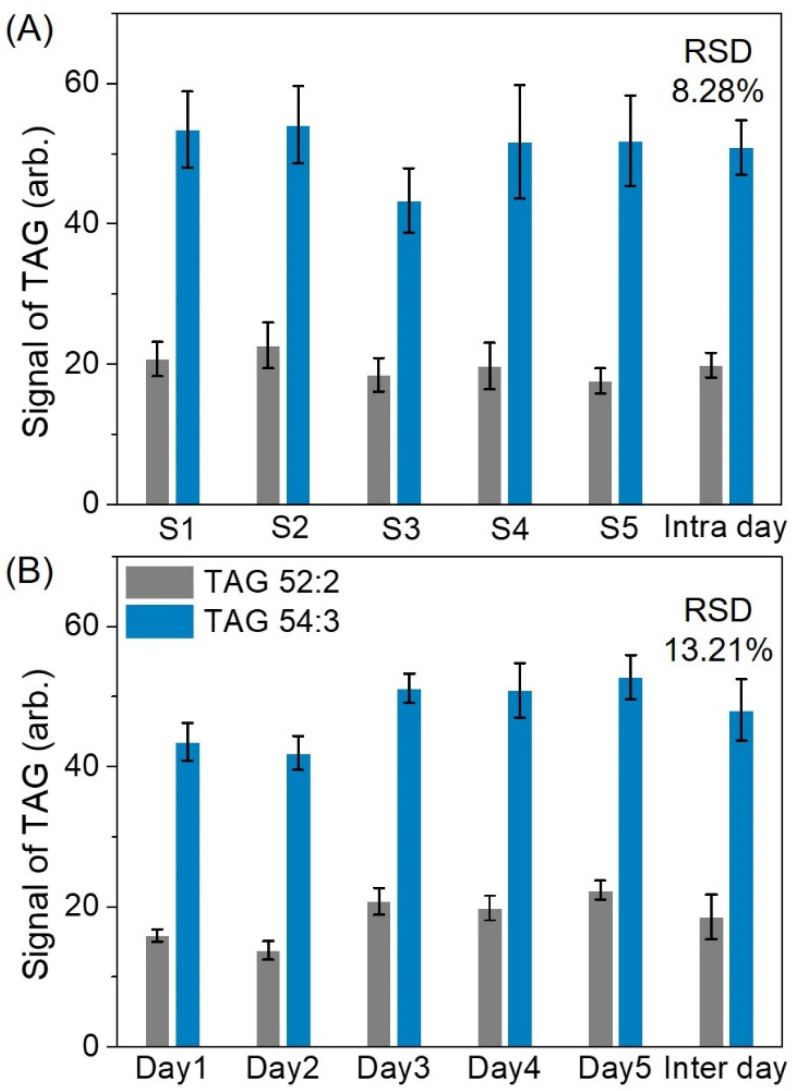
Ion intensity and error bar in the TAG signals obtained using (**A**) five samples within 1 day and (**B**) five repeated experiments in an individual day. The average intra-sample RSD was 11.95% (S1), 12.26% (S2), 11.80% (S3), 16.23% (S4), and 12.36% (S5). The average intra-day RSD was 5.82% (Day 1), 7.76% (Day 2), 6.60% (Day 3), 8.28% (Day 4), and 6.03% (Day 5). The average inter-day RSD was 13.21%. Each data bar represents the sum of 200 laser shots. Error bars represent the standard errors of 5 repeated experiments, including the average of 5 SALDI samples for each experiment, 25 spots from each sample, and 20 laser shots at each sample spot. Gray bars represent TAG 52:2—RSDs of the TAG 52:2 signal: 11.81% (S1), 14.35% (S2), 12.98% (S3), 16.81% (S4), 10.29% (S5), 5.52% (Day 1), 9.72% (Day 2), 9.18% (Day 3), 8.89% (Day 4), and 6.08% (Day 5). Blue bars represent TAG 54:3—RSDs of the TAG 54:3 signal: 10.14% (S1), 10.17% (S2), 10.62% (S3), 15.64% (S4), 12.42% (S5), 6.12% (Day 1), 5.80% (Day 2), 4.05% (Day 3), 7.66% (Day 4), and 9.20% (Day 5).

**Figure 4 molecules-26-05880-f004:**
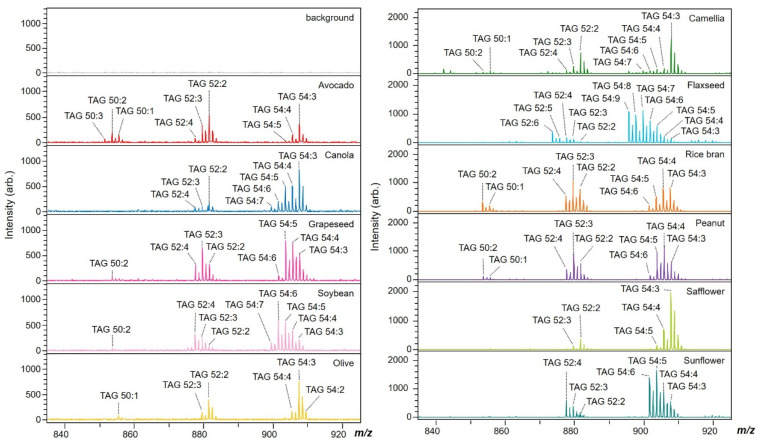
Mass spectra of TAG profiles obtained from various edible oils prepared using spray AgNPts in positive mode.

**Figure 5 molecules-26-05880-f005:**
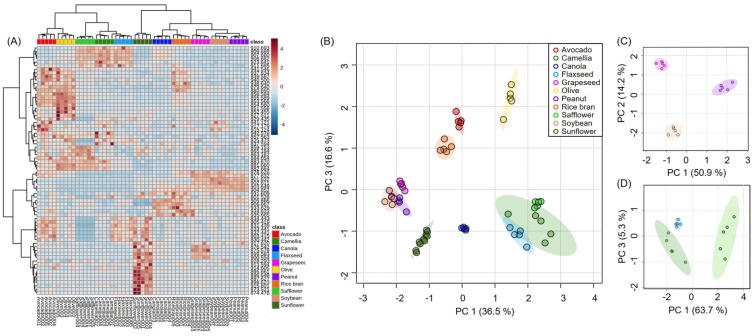
(**A**) Heatmap visualization constructed using 70 TAG features. Rows represent TAG features; columns represent samples prepared using AgNPts and the spray solution method. (**B**) PCA score plot for eleven edible oils based on the SALDI-MS results. The first and third principal components accounted for 36.5% and 16.6% of the variance, respectively. PCA score plot for (**C**) group 1 and (**D**) group 2. For group 1, the first and second principal components accounted for 50.9% and 14.2% of the variance, respectively. For group 2, the first and third principal components accounted for 63.7% and 5.3% of the variance, respectively.

**Table 1 molecules-26-05880-t001:** Division of edible oils on the basis of their SALDI mass spectra and heatmap results.

Group	Oil Species	Characteristic Peak in Spectra
1	Peanut oil and soybean oil Grapeseed oil	Abundant peak at *m*/*z* 901.6 and 903.6 Similar to soybean oil, but with an additional peak at *m*/*z* 899.6
2	Flaxseed oil and safflower oil Camellia oil	Abundant peak at *m*/*z* 905.7 Camellia oil with an additional peak at *m*/*z* 881.7
other	Avocado oil	Abundant peak at *m*/*z* 853.6, 855.6, 879.6, 881.7
	Canola oil	Abundant peak at *m*/*z* 905.7 (intensity of *m*/*z* 905.7 higher than *m*/*z* 907.7)
	Olive oil	Abundant peak at *m*/*z* 853.6, 855.6, and 881.7
	Rice bran oil	Abundant peak at *m*/*z* 879.7, 903.6, and 905.7
	Sunflower oil	Abundant peak at *m*/*z* 901.7

## Data Availability

Not applicable.

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
