# Peer review of "Utilizing AgNPt-SALDI to Classify Edible Oils by Multivariate Statistics of Triacylglycerol Profile"

_molecules, 2021, doi:10.3390/molecules26195880_

Round 1

Reviewer 1 Report

The manuscript entitled "Utilizing AgNPt-SLADI to Classify Edible Oils by Multivariate 2 Statistics of Triacylglycerol Profile" by Lee and co-workers originally reports the use of mass spectrometry (SALDI-TOF-MS) to analyze triacylglycerol TGA profiles in various commercially available edible oils.

The proposed by the authors' strategy is very important for the solution of an essential problem low reproducibility of the results of measurements TAG profile in oil using the MALDI_TOF-MS technique associated with the difficulty in obtaining a good homogeneity of the matrix/sample system.

In particular, the method developed by authors with the use of silver nanoparticles  AgNPts of an appropriate shape as an assisting material and preparation of sample by solution spray method provides the most homogenous samples. Consequently, the high reproducibility of signals of fatty acid and TGA was obtained with the applied SALDI method, where preparation of sample does not require crystallization of matrix and analyte

The experimental data presented are well done, relevant references are provided, and the narrative is concise. Overall, it is a very solid article that should improve food quality control procedures. Therefore I recommend its publication in Molecules journal in its present form.

Author Response

The authors appreciate these positive comments. 

Reviewer 2 Report

The topic of this research is very interesting and conducted with proper scientific procedures. But, the manuscript has certain major issues which need to be addressed before its final acceptance.

Observations:

In general, the manuscript needs to be thoroughly revised for linguistic errors. There are some grammatical mistakes.

Lines 2, 17, and 213:  revise and correct the abbreviation “SLADI” throughout the manuscript

Line 8:  Highlights results should be mentioned in the abstract

Line 17: revise the keywords and also capitalize the initial letter of all the keywords.

Lines 9, 14, 71, 81, 88, 101, 114, 116, 120……. etc.: It is preferable not to use personal pronouns in scientific writing except when necessary.

Lines 139 and 164: Section No. 2.2 and Section No. 2 3. have the same title. There should be specific titles according to the content or experiments. So, two different headings can be specified, or these sections can be combined under one title

Lines 99-101: Explain why a lower concentration (1.0 μl/ ml) gives stronger signals compared to higher concentrations (2 to 10 μl/ ml)?

Figure 2: The standard deviation looks very high

Line 98: It is important to mention the statistically significant differences in the results and figures

Line 203: Discussion:

1 - The discussion was not enough and did not achieve the desired goal of the results, and there is a lack of comparison between methods used and tested samples. Therefore, the authors should add more details and compare their results with other publications, explaining their findings.

2-  The mechanism behind the obtained results at some places is missing.

References 
Authors should follow the guidelines of the journal to write references throughout the manuscript.

Author Response

The authors appreciate these comments given by the Editor Professor Jasmin Gao and the reviewer for improving the quality of this manuscript. We have made changes and/or provided explanations that we hope will serve to alleviate your reservations and concerns. Please find our itemized responses and our revisions/corrections in the re-submitted files.

Reviewer 3 Report

Technologies that use nanoparticles for analysis are not new.  SALDI has been implemented and studied for analytical purposes for more than 10 years now.  In contrast, conventional laser desorption MALDI-MS is a quite mature technology for characterization and quantitative analysis of many biological molecules (peptides, proteins, metabolites and lipids). The authors aim to show that edible oils can be distinguished by SALDI-MS, a novel technique that uses Ag-doped nanoparticles as a way to photo excite triacyl glycerides.  The spectral distribution of these TAGS can then be used to test for the authenticity of edible natural oils.  The TAG profile is the basis for a multivariate statistical analysis which the authors conducted in the manuscript.

The introduction is somewhat incomplete.  The authors argue that LC-ESI-MS is inconvenient because it involves a "time consuming" solvent extraction, which was used for the characterization of olive oil varieties.  But that is not the analysis attempted here, and given that the ultimate sample preparation does involve a dilution step ahead of SALDI analysis, I can't see this technology replacing the rather important analysis of olive oil.  The authors argue that MALDI-MS is inadequate due to inconsistencies in the sample crystals.  But that is only the case when a thin layer preparation is made, but there is also dried-droplet and solution spray MALDI applications.  In fact, in Figure S2B it is clear that CHCA coupled MALDI has higher sensitivity than the 3 types of nanoparticles that were utilized herein, and the reduction in scatter as reported by the RSD values is less than 50% (62% to 41%) going from traditional MALDI to SALDI.  Hence the results shown in Figure 2 are not conclusive, since DHB, a less common sample matrix was used as a comparison, even in a case where the TGA profiles are not really much different.

Other issues that I have a problem are:

  1. The authors need to explain why the intra-sample RSD are higher by quite a bit when compared to the intra-day and inter-day RSD values (11-14% versus 5-8%) as indicated in the caption in Figure 3.
  2. Was spray of CHCA solutions used or have it been used in the context of analysis of edible oils?
  3. Are the groups derived in Table 1 constructed on the basis of the PCA analysis?
  4. What criterion did the authors use for selecting the axis (PCA vectors) in Figure 5?  My experience is that you first select the 2 highest contributors, and if required, add more, in order to observe better score plots.  But here it looks like PCA2 was ignored, which presumably accounted for somewhere between 16 and 36% of the variance.
  5. In lines 210 and 211 it should read "experiments" and "lower".  In line 220 it should be "expand"
  6. I would like to read a more detailed explanation of why the triangular shapes of the nanoparticles has a significant improvement on the ionization process. Also a better explanation of what the NP colloids are should be helpful.
  7. it is my understanding that CHCA doped MALDI samples, regardless of the method of application, benefit in ionization by addition of 0.1% formic acid to the droplets before drying.  Would that also work with the Ag-NP methods?  Please explain.
  8. In Figure 5 there are three PCA plots.  The smaller PCA figures include only the edible oils in group1 (B) and group 2 (C).  So it appears that there are only 2 types of oils in each group.  Can the authors explain why there are 3 clusters of variance instead of the expected 2 in these 2 plots?

Author Response

The authors appreciate these comments given by the Editor Professor Jasmin Gao and the reviewer for improving the quality of this manuscript. We have made changes and/or provided explanations that we hope will serve to alleviate your reservations and concerns. Please find our itemized responses and our revisions/corrections in the re-submitted file.
